# Weakly supervised deep learning model with size constraint for prostate cancer detection in multiparametric MRI and generalization to unseen domains

**Robin Trombetta**[1]                                    ROBIN.TROMBETTA@CREATIS.INSA-LYON.FR
**Olivier Rouvière**[2]                                    OLIVIER.ROUVIERE@CHU-LYON.FR
**Carole Lartizien**[1]                                    CAROLE.LARTIZIEN@CREATIS.INSA-LYON.FR

[1] *Univ. Lyon, CNRS UMR 5220, Inserm U1294, INSA Lyon, UCBL, CREATIS, France*

[2] *Hospices Civils de Lyon, Radiology Department, Edouard Herriot Hospital, Lyon, France*

**Editors:** Accepted for publication at MIDL 2024

## Abstract

Fully supervised deep models have shown promising performance for many medical segmentation tasks. Still, the deployment of these tools in clinics is limited by the very time-consuming collection of manually expert-annotated data. Moreover, most of the state-of-the-art models have been trained and validated on moderately homogeneous datasets. It is known that deep learning methods are often greatly degraded by domain or label shifts and are yet to be built in such a way as to be robust to unseen data or label distributions. In the clinical setting, this problematic is particularly relevant as the deployment institutions may have different scanners or acquisition protocols than those from which the data has been collected to train the model. In this work, we propose to address these two challenges on the detection of clinically significant prostate cancer (csPCa) from bi-parametric MRI. We evaluate the method proposed by (Kervadec et al., 2018), which introduces a size constaint loss to produce fine semantic cancer lesions segmentations from weak circle scribbles annotations. Performance of the model is based on two public (PI-CAI and Prostate158) and one private databases. First, we show that the model achieves on-par performance with strong fully supervised baseline models, both on in-distribution validation data and unseen test images. Second, we observe a performance decrease for both fully supervised and weakly supervised models when tested on unseen data domains. This confirms the crucial need for efficient domain adaptation methods if deep learning models are aimed to be deployed in a clinical environment. Finally, we show that ensemble predictions from multiple trainings increase generalization performance.

**Keywords:** Prostate cancer detection, Weakly supervised learning, Domain generalization, Multiparametric MRI, Deep learning

## 1. Introduction

Over the last years, deep learning models have become state-of-the-art methods in almost all medical imaging applications, including segmentation and detection. Among data-oriented methods, fully supervised models remain the most common and best performing ones. However, gathering numerous expert-annotated data to train such models is a very time and ressources consuming process, restraining the current use of such models in the medical field. For this reason, other promising paradigms have also been explored such as semi-, weakly- or unsupervised learning (Bosma et al., 2023; Baur et al., 2021). They aimed to mitigate the need of annotated data to train deep learning models.

Another known drawback of deep learning methods is the limited generalization capacity to unknown data distribution. It has been shown that when tested on out-of-distribution data, deep learning models can significantly underperform compared to in-distribution evaluation (Boone et al., 2023). Yet, the robustness of models to unseen domains is an absolute necessary condition for their use in clinical settings given the inherent heterogeneity among scanners and acquisition protocols between and within clinical institutions.

In this work, we propose to tackle these two problems in the challenging task of detecting and localizing clinically significant (ISUP grade group $\geq 2$) prostate cancer (csPCa) lesions in multi-parametric MRI. This is a task of primary clinical interest, as shown by the recent success in the urology community of the PI-CAI (The Prostate Imaging: Cancer IA)[1] challenge. Many recent works have tried to improve the automatisation of cancerous prostate lesions detection (Bhattacharya, 2022). Most proposed deep learning strategies focus on supervised models, with architectures such as nnUNet ranging among the top performing on the PI-CAI challenge dataset. A few recent works have proposed self- or weakly supervised approaches (Tardy and Mateus, 2021; Bateson et al., 2021), leveraging bounding boxes (Baumgartner et al., 2021), scribbles or patient-level annotations (El Jurdi et al., 2021; Yang et al., 2021), partially lowering the gap with supervised approaches.

Our contributions in this work are threefold:

- We evaluate the method proposed by (Kervadec et al., 2018) for the challenging task of segmenting csPCa lesions in multiparametric MRI and achieve performances close to strong fully supervised baselines using only circle scribbles and image-level priors.

- We evaluate how the scribble annotation process impacts performance of weakly supervised model and show that the model is robust to various weak annotation strategies.

- We evaluate the models both on in-distribution validation data and unseen test images to evaluate the drop in performance in the generalization configuration, and show that our weakly supervised model is less prone to such effect.

- We quantify to what extent ensemble predictions from multiple trainings improve generalization of deep learning models.

## 2. Material and Method

### 2.1. A weak segmentation model based on object size constraint loss function

In (Kervadec et al., 2018), the authors proposed a loss function for partially annotated data that aims to impose a size constraint on the predicted segmentations of a model. The partial cross-entropy $\mathcal{H}$, computed only on the annotated pixels $\Omega_a$, is combined with a constraint loss $\mathcal{C}$ that adds a quadratic penalty to the model on the total sum of its predictions for class $c$ if it is outside a defined range $[a, b]$. More specifically, let $V_c = \sum_{p \in \Omega} S_{p,c}$ be the sum of the probabilities $S_{p,c}$ for class $c$ of every pixel $p$ in the image domain $\Omega$. The constraint loss is given by :

$$\mathcal{C}(V_c) = \begin{cases} (V_c - a)^2 & \text{if } V_s < a \\ (V_c - b)^2 & \text{if } V_S > b \\ 0 & \text{otherwise} \end{cases} \qquad (1)$$

---

1. The PI-CAI grand challenge : https://pi-cai.grand-challenge.org/PI-CAI/

The total cost function is then defined as

$$\mathcal{H}(S) + \lambda \mathcal{C}(V_S) \tag{2}$$

where $\lambda$ is a positive constant weighting the two terms, $V_S = \sum_{p \in \Omega} S_p$ with $S_p$ the softmax probability at pixel $p$ in the image domain $\Omega$.

The size constraint loss term was initially used in a binary segmentation problem to improve prostate gland segmentation in multiparametric partially labeled MRI in (Kervadec et al., 2018). (Duran et al., 2022) extended the binary formulation to a multi-class output with $C$ classes and evaluated it on a prostate cancer detection task. They used this constraint at the image level with *image tag priors*, following the definition in Kervadec et al. (Kervadec et al., 2018), that is enforcing the presence of the target class by setting $a = 1$ and $b = |\Omega|$ (the image domain) or the absence of the target with parameters $a = b = 0$. This implementation achieved promising performance for segmentation and grading of PCa lesions in a weakly supervised setting on the Prostatex-2 challenge and a private datasets.

We extend the work of (Duran et al., 2022) on prostate cancer detection by leveraging the constraint term referred to as *common bounds* introduced by (Kervadec et al., 2018), whose principle is to introduce more precise lower and upper bounds $a$ and $b$ depending on the size of the lesions in the ground truth. These bounds are a way to introduce prior knowledge on the objects to detect to compensate for the partially labeled data. We implement this method by imposing a common bounds constraint both on the prostate class and the CS lesion class.

## 2.2. Data description

The experiments are conducted on three datasets, described hereunder :

- PI-CAI challenge public training dataset. It contains 1500 multi-parametric MRI (T2w, DWI and ADC) exams from 3 Dutch centers acquired on 7 different scanners, 5 from Siemens Healthineers and 2 from Philips Medical Systems. It includes 328 cases from the Prostate-X challenge (Armato III et al., 2018). Of all the exams available, we only use the 1295 that are manually annotated by expert clinicians, and do not leverage the 205 exams with AI-derived lesion segmentations.
- The Prostate158 (Adams et al., 2022) train and validation datasets. It consists of 139 annotated biparametric MRI (T2w, DWI) acquired at a German university hospital on 3T Magneton Vida and Skyra scanners from Siemens Healthineers.
- A private dataset, containing 219 multi-parametric MRI (T2w, DWI and ADC) exams acquired in clinical practice in two French hospitals on three different scanners : 26 exams were carried out on a 3T Ingenia scanner (Philips Medical Systems), 67 on a 1.5T Symphony scanner (Siemens Healthineers) and 126 on a 3T Discovery scanner (GE Heathcare). It was declared to the appropriate national administrative authorities (CPP L 09-04 and CNIL 08-06) and patients gave written informed consent for researchers to use their MR imaging data. All patients underwent a radical prostatectomy and prostate focal lesions manually outlined by expert radiologists on the different imaging sequences were validated against the prostatectomy gold standard ground truth.

Both T2-weigthed (T2w) and apparent diffusion coefficient (ADC) MR maps were used as input channels. The latter modality was registered to the former, all images were resampled to a $1 \times 1 \times 3$ mm$^3$ pixel size and cropped to $96 \times 96 \times 20$ volumes. Images intensities

were linearly normalized into the range [0, 1] for each patient and each modality. More details about these datasets can be found on Appendices A and B, including lesion volume distributions and histograms of intensities for T2-weighted imaging and ADC maps.

## 2.3. Weak annotations

The aim of weak annotations is to mimic what could be an easier and faster way for clinicians to provide annotations on real images. For this purpose, we replace full segmentations by circles of maximum radius of 3 mm inside each individual lesion. The centers of the circles are drawn randomly and independently on each axial slices. If the lesion is too small to fit a circle of this size, the radius is reduced until a circle can fit inside the lesion. The prostate gland is also annotated is such way, with only one circle per slice. In total, weak annotations only represent 14% of the full masks of CS lesions, considerably reducing the amount and complexity of annotations and thus the time needed for experts to make these annotations. Illustrative circle annotations are depicted on Figure 3. Appendix F evaluates how the best weakly supervised model performs when other annotation strategies are adopted.

## 2.4. Experiments

We compare several weakly supervised methods and fully supervised baseline models. For our proposed scribble based weak model, we consider two main configurations : one with partial cross-entropy (CE) and the image tag (IT) and one with partial CE, image tag and common bounds (CB) constraint loss terms. We compare them to a simpler weak model with partial cross-entropy and negative cross-entropy (denoted Partial CE) as well as to fully supervised baselines trained with cross-entropy and generalized DICE loss on the full available annotations. We use 2D and 3D MONAI's DynUNet (Cardoso et al., 2022) as backbone architectures for the proposed weak and fully supervised models. As for comparison to other weakly supervised models, we train nnDetection (Baumgartner et al., 2021) with ground truth segmentations masks being 3D rectangular cuboids framing full lesion annotations (*nnDetection full*) or weak scribble annotations (*nnDetection weak*). Note the comparison between these models and the ones with size constraints is not straightforward as they do not use the same kind of weak annotations.

All models are trained in 5-fold cross-validation on the PI-CAI dataset. They are first evaluated in the *in-distribution* setup, meaning we report the mean performance on the 5 validation folds of the PI-CAI dataset. Then, we appraise the models in the generalization setup by testing them on data from two unseen domains, namely Prostate158 and our private database. Moreover, for each method, we combine the best models of each training fold into a single ensemble model, for which the lesion probability maps are computed as the average of the probability maps of the 5 aggregated models. These ensemble models are only tested on the two unseen data domains.

## 2.5. Evaluation metrics

The models are evaluated both at lesion and patient levels. Following PI-CAI guidelines, a detection map is made of non-overlapping and non connected clusters, representing predicted csPCa lesions. Each lesion is assigned a unique probability score, chosen as the average of the probabilities of the cluster's voxels. At a lesion level, we report metrics derived from the free-response receiver operating characteristics (FROC) curve which shows sensitivity as a function of the number of false positive detections per patient. In continuity of previous works done on this csPCa detection task (Bosma et al., 2023; Saha et al., 2021),

we consider a predicted lesion as a true positive if it intersects a ground truth lesion with an intersection-over-union (IoU) ratio of at least 0.1. Since there is no consensus metric to summarize a FROC curve, we choose to report the sensitivity at 1 false positive per patient. Another complementary indicator of the performances of detection models is the average precision (AP), defined as the area under the precision-recall curve. Finally, for the patient-level diagnosis performance, the area under the ROC curve (AUROC) is reported. The patient's overall likelihood of harboring csPCa is defined as the maximum score of the predicted lesion clusters.

### 2.6. Implementation details and hyperparameters

All hyperparameters were determined with grid search on the first fold of the PI-CAI training/validation splits (see Appendix C for more details). The lower and upper bounds associated with the CS lesions class were set to 5 and 500 voxels for 2D models and to 30 and 4 000 voxels for 3D models. For the prostate class, they were set to 100 and 2 500 in the 2D case and 10 000 and 40 000 in the 3D case. The fully supervised baselines were trained with cross-entropy and generalized DICE loss. All models were trained during 200 epochs with Adam optimizer, a learning rate of $10^{-3}$ and a weight decay of $10^{-4}$. To compensate for the low amount of lesions in the PI-CAI dataset (see Table 1), sampling was weighted such that 2D transverse slices for 2D models – or 3D volumes for 3D models – with and without lesions have the same probability of being drawn in a batch. For post-processing, predicted lesions of size inferior than 15 voxels are discarded.

### 3. Results

### 3.1. Classification and detection performances

Figure 1 shows performance of all considered models for the three metrics of interest, namely sensitivity at 1 FP, AP and AUROC. Figure 3 provides examples of visual results of lesion detection maps for some 3D models. Extended visual results, including of 2D and ensemble models, are showcased in Appendix E.

First of all, it is important to note that the best performing model, namely the 3D supervised DynUNet, achieves a mean AUROC of 0.82 and mean AP of 0.42 thus producing a mean aggregated score of 0.62. This performance compares well against the best achievable reported metric on the PI-CAI challenge leader-board. We thus consider it a reliable baseline for our comparison. Surprisingly, the 3D supervised DynUNet is still outperformed by 2D models in term of sensitivity at 1 FP, including by models trained with weak labels.

The two weak models with size constraints (CE+IT, CE+IT+CB) clearly outperform the model trained only with partial and negative cross entropies (Partial CE), showing the interest of the additional size constraint cost functions. Between the image tag (IT) and the common bounds (CB) losses, the latter achieves a higher score in 22 of the 30 configurations. We use the term configuration to refer to a pairwise comparison between models with the same spatial dimension input (2D or 3D) and type of model (ensemble or not) evaluated on a given dataset and for a given metric. For instance, AUROC comparison between 2D CE+IT and 2D CE+IT+CB on Prostate158 accounts for one configuration. Remarkably, the models that have been trained with weak labels can outperform fully supervised models. They also perform favorably compared to nnDetection in most cases. The weakly supervised CE+IT+CB model achieves better scores than its supervised counterpart in almost all 2D configurations, but only 2 times out of 30 in 3D.

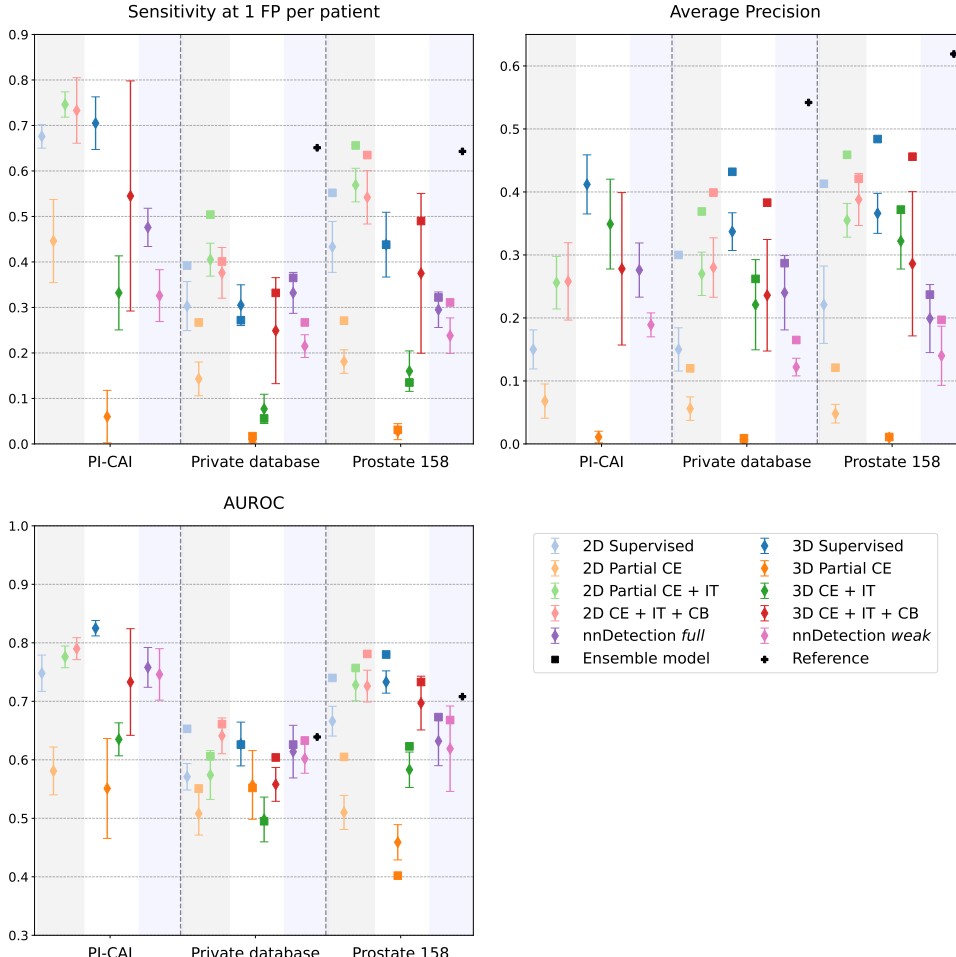

Figure 1: Classification and detection performances of all models. Reference designates fully supervised 3D DynUNet trained and tested on the Prostate158 or private dataset in 5-fold cross-validation setup. See Appendix D for detailed numerical values.

For almost all models, metrics and datasets, model ensembling improves the generalization performances, with a mean improvement of 20% among all models and metrics. We can note that for sensitivity at 1FP and AUROC, some models, especially with ensemble predictions, equal or surpass the reference metrics, which is the mean performance over 5 folds of the 3D fully supervised DynUNet trained and tested on the Prostate158 or private datasets, respectively. However, these reference models remain better in terms of AP.
Finally, the comparison study provided in Appendix F shows that the weakly supervised models are robust to several scribble annotation strategies and that the one we chose does not bias the model towards an overestimation of its performance.

## 3.2. Generalization to unseen data domains

Figure 2 shows the relative performance of the models on the two test datasets, that is Prostate158 and our private database, compared to the performance of the same model evaluated on the in-distribution validation dataset. We did not report the results for the

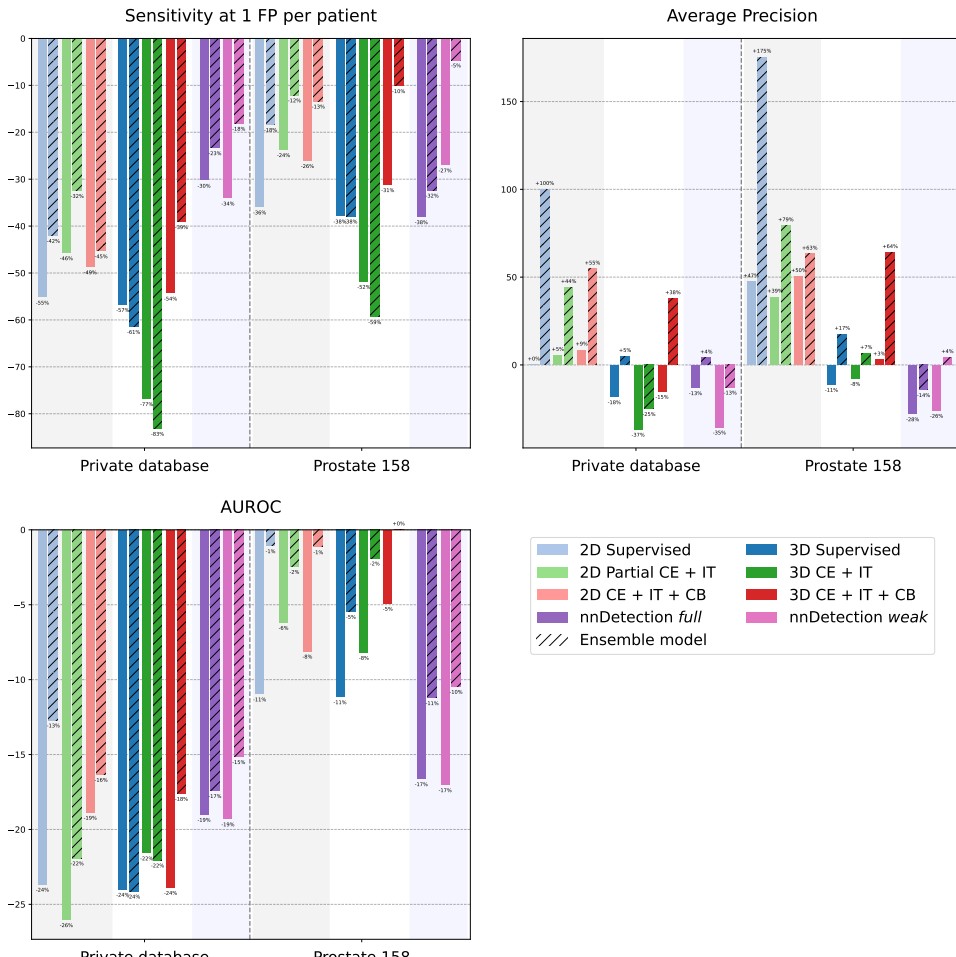

Figure 2: Relative change in performances on out-of-distribution test datasets. The reported values are the ratio between a model's performance on a test dataset (Prostate158 or our private dataset) and its cross-validation performance on PI-CAI.

cases where the models are trained with partial and negative cross entropies as their absolute performances are much lower than the others.

For most of the metrics and models, there is, as one could have expected, a notable drop in performances when the models are evaluated on a test set that has been acquired in a different setup from that of the training dataset. The average performance decrease ratio is of 28% among all models and metrics. This can reach values as low as -61% for supervised models. Quite surprisingly, the Average-Precision score is even or better – sometimes by a large amount – on the test datasets than on the validation datasets for many models.

Compared to fully supervised models, the weak models trained with CB constraint loss has a more favorable relative change in 19 configurations out of 24 (6 configurations correspond to the PI-CAI dataset and are thus not considered here). This advantage is also found when compared to the models trained with the IT loss. Ensemble predictions, with a rule as simple as averaging the output probability maps of models obtained from several trainings, almost always help reducing the performance gap in generalization.

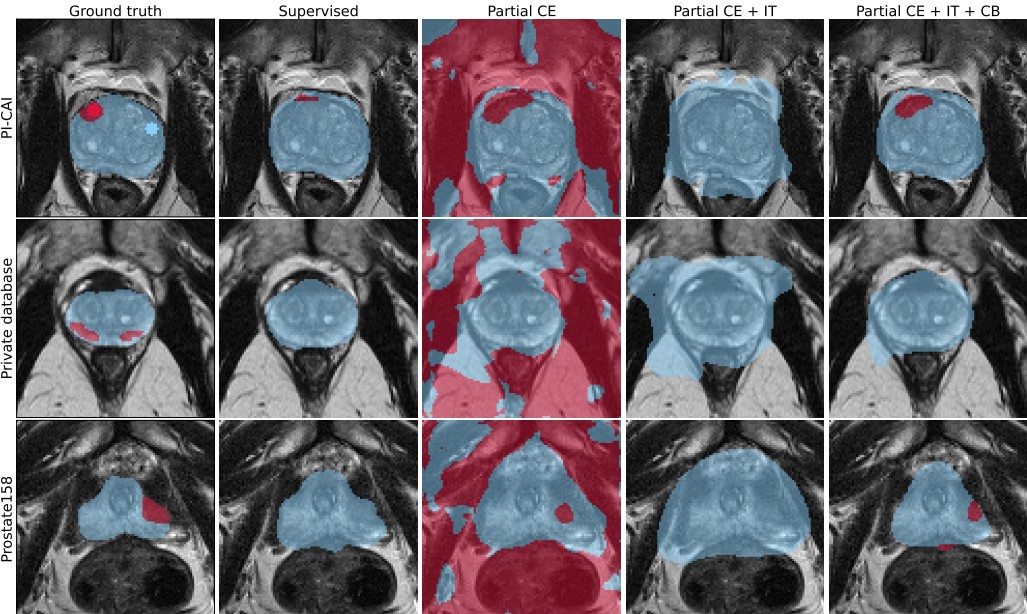

Figure 3: Example prediction maps of several 3D models. More visual results can be found in Appendix E. Blue color is for prostate and red for clinically significant lesions.

## 4. Discussion and conclusion

Our proposed weakly supervised method achieves competitive results compared to fully supervised baselines, while requiring only 14% of annotation voxels of clinically significant lesions. It consistently outperforms 2D supervised DynUNet trained with cross-entropy and generalized DICE loss. In the 3D configuration, the model trained with full segmentation annotations remains better overall. The addition of the more precise common bounds (CB) size constraint gives better results compared to the image tag (IT) model that was proposed in (Duran et al., 2022).

Among all compared methods, the weak model with CB loss is the most robust to unseen data domains. As seen on Figure 2, it indeed suffers the least from a performance drop when tested on data that do not belong to the training distribution.

Our study confirms, for the task of csPCa lesion detection and segmentation, that heterogeneity between training and test databases noticeably impacts performance of deep learning models and is thus an issue of first interest if such models are aimed to be used in a clinical environment. We show that one simple way to mitigate this issue is to make ensemble predictions from multiple trainings, as this allows decreasing the performance drop in almost all the configurations we have tested. Finally, it is important to note that the best models trained on PI-CAI reach performances on unseen domains that can be on par with fully supervised models trained on these datasets. This is an encouraging result that supports the current trend of building models on a given large training dataset, in a weakly or fully supervised setup, and deploying it on other institutions that have less or no annotated data. Further work includes refining the hyperparameters of the scribble based weak models as well as designing task-specific and few shot domain adaptation methods to better handle dataset heterogeneities.

## Acknowledgments

This work was supported by the RHU PERFUSE (ANR-17-RHUS-0006) of Université Claude Bernard Lyon 1 (UCBL), within the program "Investissements d'Avenir" operated by the French National Research Agency (ANR). It was also partly funded by France Life Imaging (grant ANR-11-INBS-0006).

This work was granted access to the HPC resources of IDRIS under the allocation 2023-AD011013971 made by GENCI.

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

# Appendix A. Lesion characteristics for each database

Table 1: Summary of positive cases and total number of lesions for each dataset.

| Database | # of positive cases / total patients | # of CS lesions |
|---|---|---|
| PI-CAI | 220 / 1295 (17%) | 301 |
| Private dataset | 183 / 219 (84%) | 408 |
| Prostate158 | 82 / 139 (59%) | 236 |

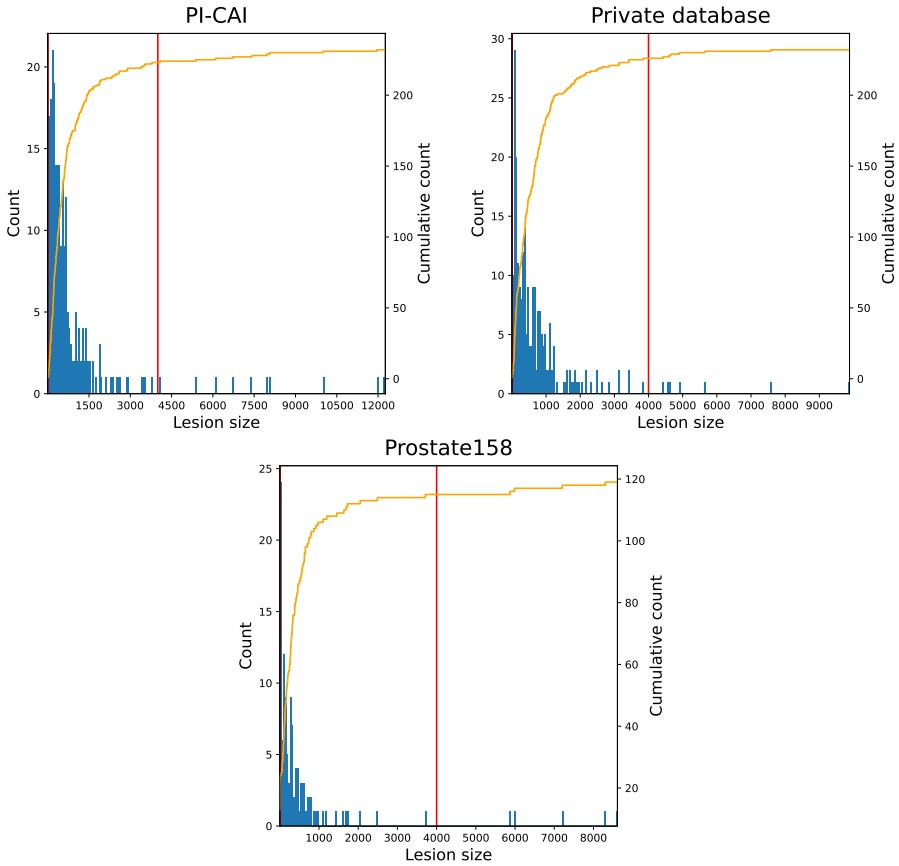

Figure 4: Histograms (blue) and cumulative histograms (orange) of lesion sizes in 3D for the three datasets. The unit of lesion sizes is the number of voxels for a volume with a spatial spacing of $1 \times 1 \times 3$ mm$^3$. The two vertical red lines show the values of the bounds $a$ and $b$ used for the CB loss (set by grid search), which are equal to 10 and 4000 respectively.

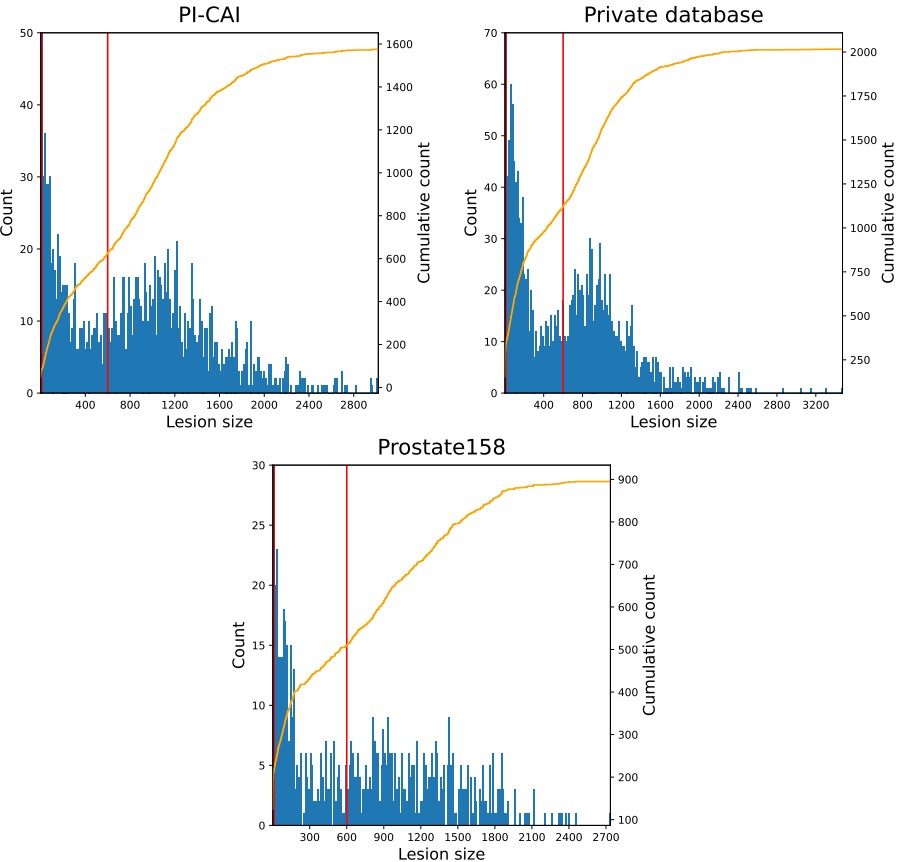

Figure 5: Histograms (blue) and cumulative histograms (orange) of slicewise lesion sizes (*i.e.* in 2D) for the three datasets. The unit of lesion sizes is the number of voxels for a volume with a spatial spacing of $1 \times 1$ mm$^2$. The two vertical red lines show the values of the bounds $a$ and $b$ used for the CB loss (set by grid search), which are equal to 10 and 600 respectively.

## Appendix B. Characteristics of MRI modalities for each database

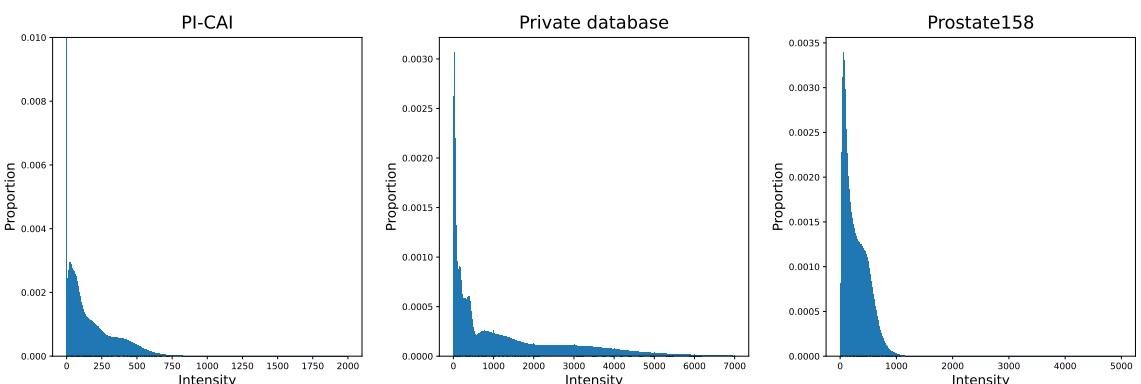

Figure 6: T2w voxel intensity distributions for each database.

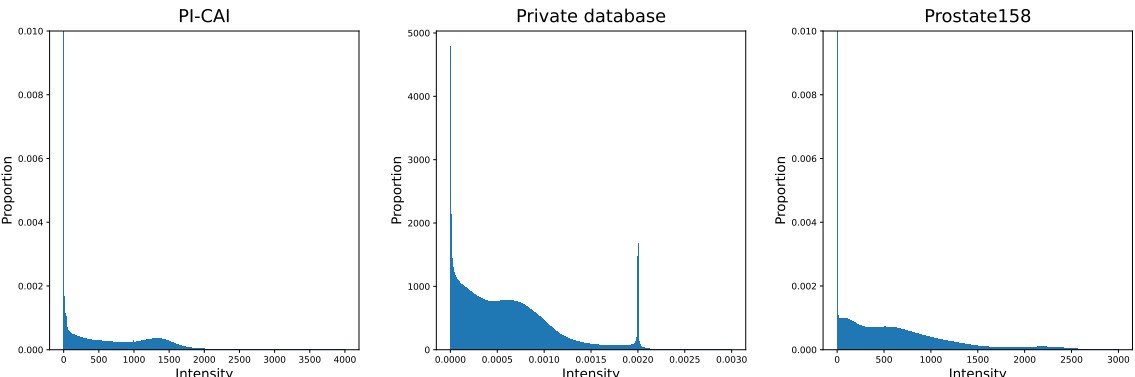

Figure 7: ADC voxel intensity distributions for each database.

## Appendix C. More details about the models

### Architecture details

The 2D DynUNet is composed of four stages with a respective number of filters of 32, 64, 128 and 256. The kernel sizes are set to 3 and the stride is of 1 for the shallowest and 2 for the others. With an input size of $[2, 96, 96]$, the deepest layers have a shape of $[256, 12, 12]$. We use Instance Normalisation layers and a dropout of ratio 0.1. The network has a total of 3.8 million (learnable) parameters.

The 3D DynUNet is composed of four stages with a respective number of filters of 32, 64, 128 and 256. The kernel sizes are set to 3 and the stride is of 1 for the shallowest and the deepest blocks and 2 for the others. With an input size of $[2, 20, 96, 96]$, the deepest layers have a shape of $[256, 5, 24, 24]$. We use Instance Normalisation layers and a dropout of ratio 0.1. The network has a total of 10.7 million (learnable) parameters.

### Details about the grid search

We did the grid search on the hyperparameters of the models as follows : for the weak constrained models, we first found the best combination of parameters for the learning rate, the weight decay and constraint weight $\lambda$ on the model with image tag. Once these parameters were set for the IT model, we reused them for the model with common bounds constraint and did the grid search for the parameters $a$ and $b$. For the supervised model and Partial CE model, we only did the grid search on the learning rate and the weight decay. The values that we tried for the hyperparameters are detailed hereunder :

- Learning rate : between $1e{-}4$ and $1e{-}2$ with a linear step of 0.5 in the logarithmic scale.
- Weight decay : between $1e{-}5$ and $1e{-}2$ with a linear step of 1 in the logarithmic scale.
- $\lambda$ : between $1e{-}5$ and $1e{-}2$ with a linear step of 1 in the logarithmic scale in the two-dimensional case and between $1e{-}5$ and $1e{-}9$ with a linear step of 1 in the logarithmic scale in the three-dimensional case. It optimal value was found to be $10^{-5}$ and $10^{-8}$ for 2D and 3D models respectively.

- $a$ and $b$ : $\{5, 10\}$ for $a$ and $\{100, 200, 300, 400, 500, 600\}$ for $b$ in the two-dimensional case and $\{10, 30, 50, 70, 100\}$ for $a$ and $\{1500, 2000, 2500, 3000, 3500, 4000, 5000, 6000\}$ for $b$ in the three-dimensional case. The grid search in the 2D case is smaller because an optimization of the hyperparameters had already been done in (Duran et al., 2022).
- the class weights $w_c$ were set to 0.14 for the prostate and 0.22 for the lesion based on empirical values reported in (Duran et al., 2022).

## Appendix D. Full numerical results associated to Figure 1

Table 2: Full results of the models on PI-CAI dataset. For each metric, the best model is in **bold** and the second best is underlined. † is used when a model does not reach 1 FP per patient; in such case, we report the maximum sensitivity.

| Model | Sensi at 1 FP | Average Precision | AUROC |
|---|---|---|---|
| 2D Supervised | $0.676 \pm 0.026$ | $0.150 \pm 0.031$ | $0.748 \pm 0.031$ |
| 2D Partial CE | $0.446 \pm 0.091$ | $0.068 \pm 0.027$ | $0.581 \pm 0.041$ |
| 2D CE + IT | $\mathbf{0.746} \pm 0.028$ | $0.256 \pm 0.042$ | $0.776 \pm 0.019$ |
| 2D CE + IT + CB | $\underline{0.733} \pm 0.072$ | $0.258 \pm 0.061$ | $\underline{0.790} \pm 0.019$ |
| 3D Supervised | $0.705 \pm 0.058^{\dagger}$ | $\mathbf{0.412} \pm 0.047$ | $\mathbf{0.825} \pm 0.013$ |
| 3D Partial CE | $0.060 \pm 0.058^{\dagger}$ | $0.011 \pm 0.009$ | $0.551 \pm 0.085$ |
| 3D CE + IT | $0.332 \pm 0.081^{\dagger}$ | $\underline{0.349} \pm 0.071$ | $0.635 \pm 0.028$ |
| 3D CE + IT + CB | $0.545 \pm 0.252^{\dagger}$ | $0.278 \pm 0.121$ | $0.733 \pm 0.091$ |
| nnDetection *full* | $0.332 \pm 0.081$ | $\underline{0.349} \pm 0.071$ | $0.635 \pm 0.028$ |
| nnDetection *weak* | $0.545 \pm 0.252$ | $0.278 \pm 0.121$ | $0.733 \pm 0.091$ |

| Model | Maximum sensitivity | Avg. FP per patient |
|---|---|---|
| 2D Supervised | $\mathbf{0.803} \pm 0.075$ | $3.21 \pm 0.52$ |
| 2D Partial CE | $0.339 \pm 0.071$ | $10.69 \pm 2.53$ |
| 2D CE + IT | $0.516 \pm 0.083$ | $3.67 \pm 1.06$ |
| 2D CE + IT + CB | $\underline{0.756} \pm 0.096$ | $1.55 \pm 0.52$ |
| 3D Supervised | $0.705 \pm 0.058$ | $\underline{0.69} \pm 0.15$ |
| 3D Partial CE | $0.126 \pm 0.049$ | $5.02 \pm 2.54$ |
| 3D CE + IT | $0.332 \pm 0.081$ | $\mathbf{0.12} \pm 0.04$ |
| 3D CE + IT + CB | $0.715 \pm 0.075$ | $1.42 \pm 0.85$ |

Table 3: Full results of the models on our private dataset. The results after ensembling is shown between brackets. For each metric, the best model is in **bold** and the second best is underlined (excluding the reference model). † is used when a model does not reach 1 FP per patient; in such case, we report the maximum sensitivity.

| Model | Sensi at 1 FP | Average Precision | AUROC |
|---|---|---|---|
| 2D Supervised | $0.303 \pm 0.054 \ (0.392)$ | $0.150 \pm 0.034 \ (0.0.300)$ | $0.571 \pm 0.023 \ (\underline{0.653})$ |
| 2D Partial CE | $0.143 \pm 0.037 \ (0.267)$ | $0.056 \pm 0.019 \ (0.120)$ | $0.508 \pm 0.037 \ (0.551)$ |
| 2D CE + IT | $\mathbf{0.405} \pm 0.036 \ (\mathbf{0.504})$ | $0.270 \pm 0.034 \ (0.369)$ | $0.574 \pm 0.042 \ (0.606)$ |
| 2D CE + IT + CB | $\underline{0.376} \pm 0.056 \ (\underline{0.401})$ | $\underline{0.280} \pm 0.047 \ (\underline{0.399})$ | $\mathbf{0.641} \pm 0.031 \ (\mathbf{0.661})$ |
| 3D Supervised | $0.305 \pm 0.045^{\dagger} \ (0.272)$ | $\mathbf{0.337} \pm 0.030 \ (\mathbf{0.432})$ | $\underline{0.627} \pm 0.037 \ (0.626)$ |
| 3D Partial CE | $0.007 \pm 0.003^{\dagger} \ (0.017)$ | $0.005 \pm 0.002 \ (0.009)$ | $0.557 \pm 0.059 \ (0.552)$ |
| 3D CE + IT | $0.077 \pm 0.032^{\dagger} \ (0.056)$ | $0.221 \pm 0.072 \ (0.262)$ | $0.498 \pm 0.038 \ (0.495)$ |
| 3D CE + IT + CB | $0.249 \pm 0.117^{\dagger} \ (0.332)$ | $0.236 \pm 0.089 \ (0.383)$ | $0.558 \pm 0.029 \ (0.604)$ |
| nnDetection *full* | $0.332 \pm 0.081$ | $\underline{0.349} \pm 0.071$ | $0.635 \pm 0.028$ |
| nnDetection *weak* | $0.545 \pm 0.252$ | $0.278 \pm 0.121$ | $0.733 \pm 0.091$ |
| Reference | $0.651$ | $0.542$ | $0.639$ |

| Model | Maximum sensitivity | Avg. FP per patient |
|---|---|---|
| 2D Supervised | $\underline{0.407} \pm 0.060$ (0.397) | $3.84 \pm 0.78$ (1.47) |
| 2D Partial CE | $0.335 \pm 0.051$ (0.371) | $11.00 \pm 2.76$ (8.23) |
| 2D CE + IT | $\mathbf{0.512} \pm 0.046$ (**0.522**) | $3.71 \pm 0.89$ (1.84) |
| 2D CE + IT + CB | $0.391 \pm 0.073$ ($\underline{0.401}$) | $1.55 \pm 0.32$ (0.61) |
| 3D Supervised | $0.306 \pm 0.045$ (0.272) | $\underline{0.73} \pm 0.25$ ($\underline{0.22}$) |
| 3D Partial CE | $0.039 \pm 0.013$ (0.034) | $4.99 \pm 2.65$ (3.86) |
| 3D CE + IT | $0.077 \pm 0.032$ (0.056) | $\mathbf{0.13} \pm 0.055$ (**0.06**) |
| 3D CE + IT + CB | $0.331 \pm 0.042$ (0.332) | $1.78 \pm 0.92$ (0.52) |
| Reference | $0.655 \pm 0.063$ | $1.40 \pm 0.20$ |

Table 4: Full results of the models on Prostate158. The results after ensembling is shown between brackets. For each metric, the best model is in **bold** and the second best is underlined (excluding the reference model). $^\dagger$ is used when a model does not reach 1 FP per patient; in such case, we report the maximum sensitivity.

| Model | Sensi at 1 FP | Average Precision | AUROC |
|---|---|---|---|
| 2D Supervised | $0.433 \pm 0.056$ (0.552) | $0.221 \pm 0.061$ (0.412) | $0.666 \pm 0.025$ (0.740) |
| 2D Partial CE | $0.181 \pm 0.026$ (0.271) | $0.048 \pm 0.015$ (0.121) | $0.510 \pm 0.029$ (0.605) |
| 2D CE + IT | $\mathbf{0.569} \pm 0.037$ (**0.656**) | $0.355 \pm 0.027$ (0.459) | $\underline{0.728} \pm 0.027$ (0.757) |
| 2D CE + IT + CB | $\underline{0.542} \pm 0.059$ ($\underline{0.635}$) | $\mathbf{0.388} \pm 0.041$ ($\underline{0.421}$) | $0.726 \pm 0.028$ (**0.781**) |
| 3D Supervised | $0.438 \pm 0.071^\dagger$ (0.438) | $\underline{0.366} \pm 0.031$ (**0.484**) | $\mathbf{0.733} \pm 0.019$ ($\underline{0.780}$) |
| 3D Partial CE | $0.027 \pm 0.018^\dagger$ (0.031) | $0.010 \pm 0.004$ (0.011) | $0.459 \pm 0.030$ (0.402) |
| 3D CE + IT | $0.160 \pm 0.044^\dagger$ (0.135) | $0.322 \pm 0.044$ (0.373) | $0.583 \pm 0.030$ (0.623) |
| 3D CE + IT + CB | $0.375 \pm 0.176^\dagger$ (0.490) | $0.286 \pm 0.115$ (0.456) | $0.697 \pm 0.046$ (0.733) |
| nnDetection *full* | $0.332 \pm 0.081$ | $\underline{0.349} \pm 0.071$ | $0.635 \pm 0.028$ |
| nnDetection *weak* | $0.545 \pm 0.252$ | $0.278 \pm 0.121$ | $0.733 \pm 0.091$ |
| Reference | $0.643$ | $0.619$ | $0.708$ |

| Model | Maximum sensitivity | Avg. FP per patient |
|---|---|---|
| 2D Supervised | $0.594 \pm 0.055$ (0.552) | $4.47 \pm 0.90$ (1.73) |
| 2D Partial CE | $0.290 \pm 0.051$ (0.323) | $8.72 \pm 1.86$ (6.03) |
| 2D CE + IT | $\mathbf{0.710} \pm 0.029$ (**0.698**) | $5.40 \pm 0.92$ (3.10) |
| 2D CE + IT + CB | $\underline{0.602} \pm 0.074$ ($\underline{0.635}$) | $2.47 \pm 0.79$ (1.05) |
| 3D Supervised | $0.452 \pm 0.082$ (0.438) | $\underline{0.94} \pm 0.24$ ($\underline{0.25}$) |
| 3D Partial CE | $0.087 \pm 0.036$ (0.094) | $5.02 \pm 2.60$ (3.78) |
| 3D CE + IT | $0.160 \pm 0.044$ (0.135) | $\mathbf{0.15} \pm 0.07$ (**0.07**) |
| 3D CE + IT + CB | $0.496 \pm 0.043$ (0.490) | $2.07 \pm 1.00$ (0.68) |
| Reference | $0.643$ | $1.05$ |

## Appendix E. Supplementary visual results

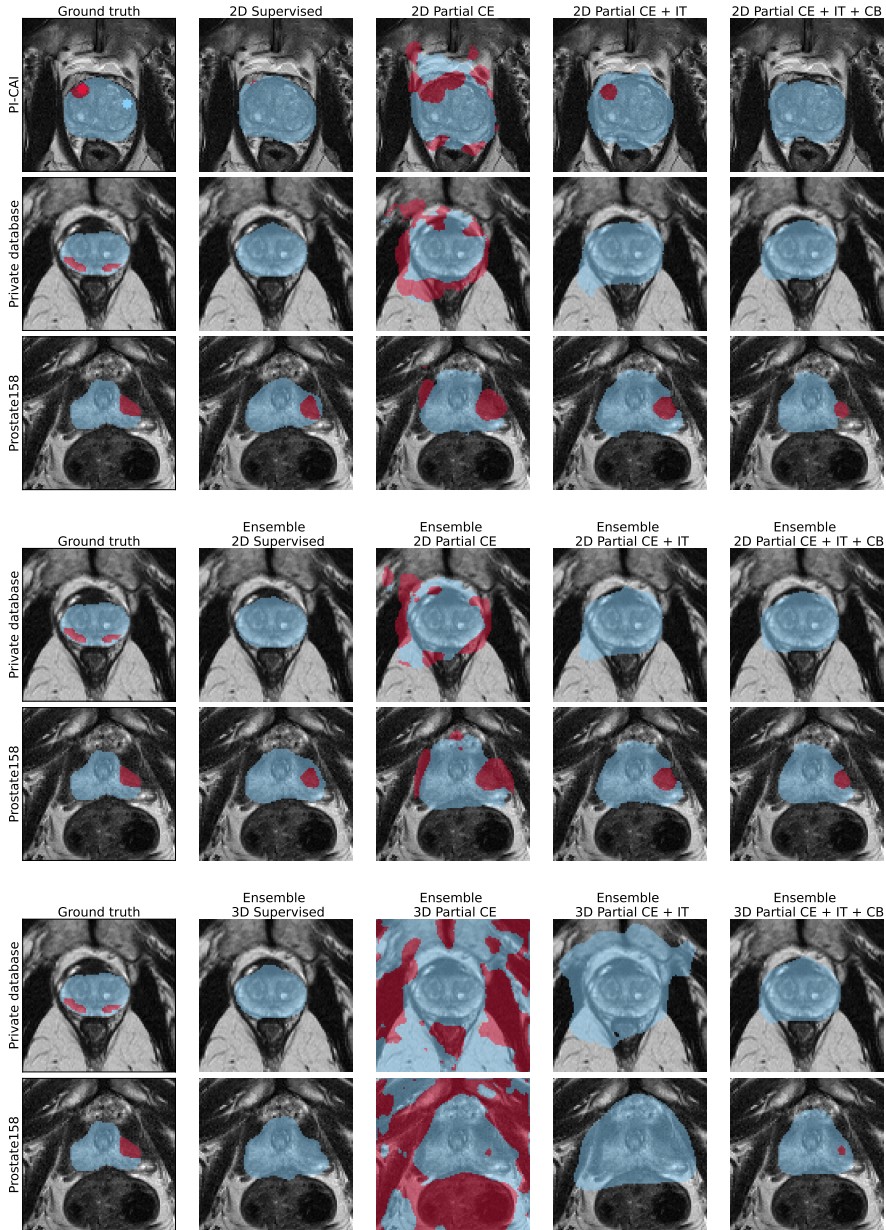

Figure 8: Examples prediction maps of several 2D models, ensemble of 2D models and ensemble of 3D models. Blue color is for prostate and red for clinically significant lesions.

## Appendix F. Study on the method for generating weak annotations

Modeling the process of obtaining the weak scribble annotations is crucial for correctly evaluating the relevance of the weakly supervised models, as an unrealistic modeling could lead to under- or over-estimated performance of the model. In order to assess the robustness of the weakly supervised models to such annotation process, we provide here comparison between several annotation methods that we consider to be realistic ways of producing scribble annotations. Note that these annotation methods apply only to the lesion class; for the prostate class the method described described in section 2.3 and referred to as *random valid* is systematically used. The methods are as follows :

- *Random valid* denotes the method described in section 2.3.
- *Center distance map.* For each lesion, we compute the Euclidean distance between each non-zero pixel in lesion mask and the nearest zero pixel. The center of the lesion is the maximum value of this map and the circular annotation that is drawn is the largest circle of radius inferior or equal to 3mm that can fit in the CS lesion mask. The amount of annotated pixels with this method is similar to the one of *random valid* (14%).
- *Random distance map.* From the distance map obtained as described above, we randomly draw a center for the circular scribble with a probability density proportional to the distance map, to bias selection of the scribble center towards the center of the lesion. In this case we draw a circle which always has a radius of 3mm, resulting in an amount of annotated pixels superior to the two previous methods (16.5 %).
- *Erosions.* For each lesion, we iteratively trim its mask to reduce its size until we obtain a surface below a certain value. We then threshold surface is equivalent to a circle of radius 3mm, it results in an amount of annotated pixels below the other methods (10%). Hence, for better comparison, we also perform such procedure with a higher threshold to get the same amount of CS lesion annotations (14%).

Table 5: Results of 2D CE + IT + CB on PI-CAI dataset for several weak annotations methods. For each metric, the best model is in **bold** and the second best is underlined.

| Model | Sensi at 1 FP | Average Precision | AUROC |
|---|---|---|---|
| Random valid | $0.542 \pm 0.059$ | $\mathbf{0.388} \pm 0.041$ | $0.726 \pm 0.028$ |
| Center distance map | $\underline{0.734} \pm 0.080$ | $0.250 \pm 0.037$ | $\mathbf{0.788} \pm 0.040$ |
| Random distance map | $0.689 \pm 0.066$ | $\underline{0.296} \pm 0.031$ | $0.643 \pm 0.032$ |
| Erosions (10%) | $0.681 \pm 0.045$ | $0.185 \pm 0.016$ | $0.759 \pm 0.019$ |
| Erosions (14%) | $\mathbf{0.746} \pm 0.062$ | $0.253 \pm 0.059$ | $\underline{0.773} \pm 0.022$ |

Table 6: Results of 2D CE + IT + CB on our private dataset for several weak annotations methods. The results after ensembling is shown between brackets. For each metric, the best model is in **bold** and the second best is underlined.

| Model | Sensi at 1 FP | Average Precision | AUROC |
|---|---|---|---|
| Random valid | $0.376 \pm 0.056 \ (0.401)$ | $0.280 \pm 0.047 \ (0.399)$ | $\underline{0.641} \pm 0.031 \ (0.661)$ |
| Center distance map | $\mathbf{0.422} \pm 0.025 \ (\mathbf{0.453})$ | $\mathbf{0.296} \pm 0.031 \ (\mathbf{0.415})$ | $\mathbf{0.643} \pm 0.032 \ (\mathbf{0.685})$ |
| Random distance map | $0.347 \pm 0.040 \ (0.371)$ | $0.247 \pm 0.066 \ (0.379)$ | $0.557 \pm 0.038 \ (0.627)$ |
| Erosions (10%) | $0.366 \pm 0.030 \ (\underline{0.431})$ | $0.251 \pm 0.023 \ (0.370)$ | $0.622 \pm 0.026 \ (\underline{0.682})$ |
| Erosions (14%) | $\underline{0.411} \pm 0.047 \ (0.426)$ | $\underline{0.288} \pm 0.042 \ (\underline{0.402})$ | $0.611 \pm 0.064 \ (0.663)$ |

Table 7: Results of 2D CE + IT + CB on Prostate158 for several weak annotations methods. The results after ensembling is shown between brackets. For each metric, the best model is in **bold** and the second best is underlined.

| Model | Sensi at 1 FP | Average Precision | AUROC |
|---|---|---|---|
| Random valid | $0.542 \pm 0.059$ (**0.635**) | **0.388** $\pm 0.041$ (0.421) | $0.726 \pm 0.028$ (0.781) |
| Center distance map | **0.564** $\pm 0.032$ (0.615) | $\underline{0.355} \pm 0.028$ (0.490) | **0.760** $\pm 0.035$ (\underline{0.788}) |
| Random distance map | $0.485 \pm 0.044$ (0.615) | $0.268 \pm 0.052$ (\underline{0.502}) | $0.684 \pm 0.037$ (0.735) |
| Erosions (10%) | $\underline{0.550} \pm 0.026$ (\underline{0.625}) | $0.346 \pm 0.023$ (**0.504**) | $\underline{0.735} \pm 0.054$ (0.773) |
| Erosions (14%) | $0.519 \pm 0.030$ (0.614) | $0.317 \pm 0.047$ (0.483) | $0.731 \pm 0.025$ (**0.808**) |

## Appendix G. Model results for prostate segmentation

Although not within the scope of this work, we provide hereunder a similar visualization than Figure 1 for the prostate segmentation Dice score. As suggested by the examples predictions maps shown in Figures 3 and 8, the regularization induced by the common bounds loss greatly improves the segmentation in the weakly supervised setup compared to models with partial cross-entropy (CE) or cross-entropy and image tag (CE+IT). Note that the bounds $a$ and $b$ were set roughly for the prostate class : in the 2D case, we used the ones that were set in (Duran et al., 2022), and in 3D case, we empirically set them based on the analysis of the distribution of prostate sizes.

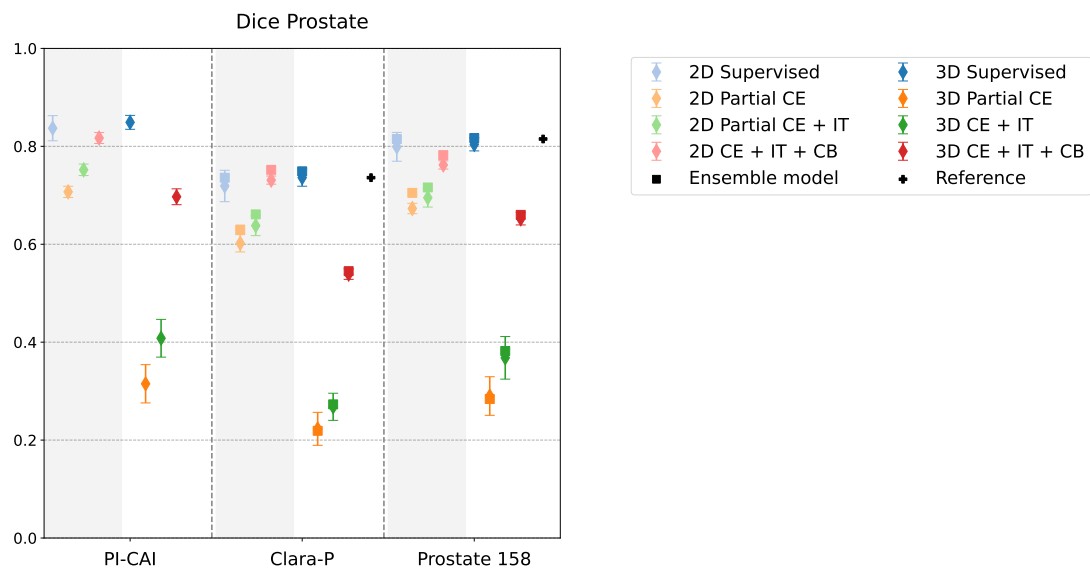

Figure 9: DICE prostate segmentation score for all models. Weak labels only represent about 2% of the total annotated voxels for the prostate class. Reference designates fully supervised 3D DynUNet trained and tested on the Prostate158 or private dataset in 5-fold cross-validation setup.

