# OpenReview forum: "Weakly supervised deep learning model with size constraint for prostate cancer detection in multiparametric MRI and generalization to unseen domains"
_MIDL.io/2024/Conference — MIDL 2024 Poster_

### Official Review · Reviewer_aYxy · 2024-02-25

**Confidence:** 4
**Preliminary Rating:** 2
**Final Rating:** 2.5

**Summary:**

The paper presents a weakly supervised method with size constraints for the detection of prostate cancer lesions is MRI. Similar to (Kervadec et al., 2018), the proposed method combines a partial cross-entropy loss on scribble-annotated pixels with a size constraint loss on the predicted foreground. The size constraint is defined with fixed lower and upper bounds on the size, and is implemented with an L2 loss as in (Kervadec et al., 2018). While the previous work in (Kervadec et al., 2018) tackled the problem of segmentation, this paper instead focuses on detecting lesions (a lesion is detected if it has a sufficient overlap with a true lesion). The model is trained on the PI-CAI dataset and tested on hold-out samples of this dataset as well as on two different datasets, Clara-P and Prostate158, verifying its cross-dataset generalization ability. Results highlight the benefit of the size loss and the difficulty of generalizing to unseen datasets.

**Strengths:**

* The paper is well written and presents clearly the proposed method and experiments.

* The cross-dataset analysis is interesting and underlines the sensitivity of deep learning models to distribution shifts in the data.

**Weaknesses:**

* The main weakness of the paper is its lack of methodological novelty. The proposed model is same as (Kervadec et al., 2018) and the overall method mainly differs by how the predicted segmentation is used (separate connected components are treated as different lesions)

* Experiments only compare the proposed method against variants (without the image tags or size losses, 2D vs 3D segmentation)

**Detailed Comments:**

(See weaknesses for other comments)

* p1: have becoming --> have become

* p1: the most common and performing ones --> the most common and best performing ones

* p2: "We propose a weakly supervised model, trained only on circle scribbles". Explain how this is a novel contribution compared to (Kervadec et al., 2018)

* p3: "We extend this work by setting more precise lower and upper bounds a and b depending of the size of the lesions in the ground truth." Explain why this is an important contribution.

* p4: "For this purpose, we replace full segmentations by circles of maximum radius of 3 mm inside each individual lesion. The centers of the circles are drawn randomly and independently on each axial slices." This scenario is not very realistic as human annotators would probably put the circle near the center of the lesion on each slice. If the segmentation is performed in 3D, having the circles placed randomly may actually offer more information as it covers a greater extent of the lesion. An alternative strategy to define the annotations is to erode the lesion mask until the target size is reached.

* p4: "Illustrative circle annotations are depicted on Figure 3." I do not see those annotations on Figure 3.

* p5: "The constraint loss weight λ was set to 10−5 and 10−8 for 2D and 3D models respectively" Why such small values?

* p5: "The weakly supervised CE+IT+CB model achieves better scores than its supervised counterpart in almost all 2D configurations, but only 2 times out of 30 in 3D." That seems normal since the size bounds are less informative as we increase the number of dimensions.

* p6 and p8 : "Figure 9 shows" . Figure 3 ?

* p6: "Quite surprisingly, the Average-Precision score is even or better – sometimes by a large amount – on the test datasets than on the validation datasets for many models." Can you explain this result?

**Justification Of Final Rating:**

Authors have provided detailed answers to my questions, however the main issues with the paper remain: lack of methodological novelty and poor evaluation of the proposed method. Nevertheless, I raise my score to Borderline reject.

**Justification Of The Preliminary Rating:**

The score can be justified by the weaknesses of the paper:

* Lack of methodological novelty. The proposed model is same as (Kervadec et al., 2018) and the overall method mainly differs by how the predicted segmentation is used (separate connected components are treated as different lesions)

* Weak evaluation. Experiments only compare the proposed method against variants (without the image tags or size losses, 2D vs 3D segmentation)

**Questions To Address In The Rebuttal:**

The authors need to clarify the novel contributions of their method and compare it against recent approaches for weakly-supervised segmentation/detection.

---

> ### Author Response · Authors · 2024-03-18
> **Response to Reviewer aYxy**
>
> Thank you for your interest in our work and your comments, Reviewer aYxy. We are glad you appreciate the work we put to evaluate the sensitivity of the models to distribution shifts. we have corrected the typos you pointed out and we hope to address your concerns below.
>
> ***1)The authors need to clarify the novel contributions of their method and compare it against recent approaches for weakly-supervised segmentation /detection.***
>
> This important point was also raised by the other reviewers. We extensively discuss it in our *reply 1) to Reviewer heuH*, and would ask Reviewer aYxy to please refer to this response.
>
> ***2)”For this purpose, we replace full segmentations by circles of maximum radius of 3 mm inside each individual lesion. The centers of the circles are drawn randomly and independently on each axial slices.” This scenario is not very realistic as human annotators would probably put the circle near the center of the lesion on each slice. If the segmentation is performed in 3D, having the circles placed randomly may actually offer more information as it covers a greater extent of the lesion. An alternative strategy to define the annotations is to erode the lesion mask until the target size is reached.***
>
> We would like to thank the reviewer for raising this point about how we characterize the placement of weak annoatations, which is crucial to justifying the relevance of our results. As suggested, we have added an appendix that compares several annotations methods. Among the methods compared, no one clearly stands out as superior to the others. Importantly, it shows that the way we initially choose to model the weak annotations does not tend to overestimate the performance of the weakly supervised models compared to other weak annotations types.
>
> ***3)”p6: ”Quite surprisingly, the Average-Precision score is even or better – sometimes by a large amount – on the test datasets than on the validation  atasets for many models.” Can you explain this result?***
>
> We do not think there is an obvious explanation for this phenomenon, but a possible explanation, at least for our private database could be found in the characteristics of our dataset itself, especially with regards to the patient inclusion criterion. Indeed, our database was built up from patients who had all undergone radical prostatectomy after the presence of cancer had been proven by biopsies. This inclusion criterion, which is more restrictive than that of the PI-CAI training dataset, results in a much higher rate of positive patients and a much higher number of lesions per patient (see Table 1 in Appendix A for details). We hypothesize that the lesions in the private database are more advanced and more easily distinguishable than those in the PI-CAI training database, which would explain the higher AP. One result that would tend to validate our interpretation is that the reference model also has a much higher AP than its counterpart trained and tested on PI-CAI (see Figure 1 and Table 2).
>
> ***4)Weak evaluation. Experiments only compare the proposed method against variants (without the image tags or size losses, 2D vs 3D segmentation)***
>
> This point was also discussed in detail in our response both to reviewers heuH and 49yd. For greater clarity and to avoid unnecessary repetition, we kindly ask reviewer aYxy to refer to the *answer 2) to reviewer heuH and answer 1) to reviewer 49yd*.
>
> ***5)”The constraint loss weight lambda was set to 10e-5 and 10e-8 for 2D and 3D models respectively” Why such small values?***
>
> The constraint cost function is not normalized and is therefore typically of the order of magnitude of the square of the objects to be detected. Please note that this explains why the lambda factor is higher in 3D than in 2D models. We could have worked out a way to normalize this cost function to avoid having to use a lambda learning factor, but the grid search to determine a relevant value for lambda was easy to conduct so we chose to preceed this way.

---

> > ### Comment · Reviewer_aYxy · 2024-03-25
> > **Thanks for the detailed answers**
> >
> > I thank the authors for answering my questions. I believe the paper has merits, however I still feel that applying an existing method to a different problem offers a limited contribution to the field. Also, the lack of comparison to weakly-supervised baselines remains an issue.

---

### Official Review · Reviewer_49yd · 2024-02-28

**Confidence:** 5
**Preliminary Rating:** 4
**Recommendation:** Poster

**Summary:**

The study addresses challenges in deploying fully supervised deep models for medical segmentation due to the time-consuming collection of expert-annotated data and issues with domain shifts. It proposes a weakly supervised method using scribble annotations to detect clinically significant prostate cancer from MRI scans. The model achieves comparable performance to fully supervised models on validation and unseen test data but experiences decreased performance on unseen data domains, highlighting the need for domain adaptation methods in clinical deployment. Ensemble predictions from multiple trainings enhance generalization performance.

**Strengths:**

- The study utilizes three datasets: the PI-CAI challenge public dataset with 1295 manually annotated MRI exams from Dutch centers, the Prostate158 dataset with 139 annotated MRI exams from a German university hospital, and a private dataset with 219 MRI exams from French hospitals, validated against prostatectomy gold standard ground truth.

- The 3D supervised DynUNet achieves a mean AUROC of 0.82 and mean AP of 0.42, comparable to the best reported metric on the PI-CAI challenge leaderboard.
Surprisingly, 2D models outperform the 3D supervised DynUNet in sensitivity at 1 FP, even models trained with weak labels.

- Models with size constraints (CE+IT, CE+IT+CB) outperform those with only partial and negative cross-entropies (Partial CE), with CB showing higher scores in most configurations.

- Weakly supervised models, especially CE+IT+CB, can outperform fully supervised ones, particularly in 2D configurations.

- Model ensembling consistently improves generalization performances across most models, metrics, and datasets, with a mean improvement of 20%.

- Evaluation metrics involve both lesion and patient level.

**Weaknesses:**

- no state of the art methods segmentation techniques employed.
-  statistically signify CSE of models not reported e.g CI
- ensemble method was introduced to reduce performance gap which underscores other models presented; more detail on how models were ensembled.

**Detailed Comments:**

Introduction of state-of-the-art segmentation techniques was lacking in the evaluation.
Statistical significance, such as confidence intervals, of model performance was not reported.
The introduction of an ensemble method aimed to reduce the performance gap, highlighting potential improvements for other presented models.

**Justification Of The Preliminary Rating:**

The evaluation did not incorporate state-of-the-art segmentation techniques.

Statistical significance, including confidence intervals, was not included in the analysis of model performance.

The introduction of an ensemble method aimed to narrow the performance gap, indicating possible enhancements for other models presented.

Regarding the better performance of 2D over 3D models, there is no clear justification for why the 3D supervised DynUNet is consistently outperformed by 2D models in terms of sensitivity at 1 false positive.

**Questions To Address In The Rebuttal:**

Why is 2d performing better than 3d is there a justification?
“Surprisingly, the 3D supervised DynUNet is still outperformed by 2D models in term of sensitivity at 1 FP, including by models trained with weak labels.”
What is the significance of introducing ensemble method, how were the models ensembled?

**Special Issue:**

No

---

> ### Author Response · Authors · 2024-03-18
> **Response to Reviewer 49yd**
>
> Thank you for your interest in our work and your comments, Reviewer 49yd, we hope to address your concerns below.
>
> ***1)no state of the art methods segmentation techniques employed.***
>
> We believe that the supervised models we are comparing with consitute solid baselines *for an evaluation with single models or simple ensembling*. The best-performing models in the PI-CAI challenge are typically designed to win such challenges, i.e. a clever ensembling of well-established models, such as nnUnet and nnDetection in this case, to improve the robustness of the predictions. For further details on the limits of comparing our models with those of the PI-CAI ranking and with the winning methods of the challenge, *please refer to the answer 2) given to reviewer heuH*.
>
> As stated in the global response, we are currently training an nnDetection model on PI-CAI data and aim to include its results in the next few days, to provide for readers a comparison with a state-of-the-art model in weakly supervised segmentation. However, please note that that nnDetection is supervised by bounding boxes annotations, which we believe to be stronger ground truths than dots when models are evaluated with detection metrics.
>
> ***2)statistically signify CSE of models not reported e.g CI***
>
> As all the metrics are patient-level or lesion-level, they have to be computed on a whole dataset. Hence, to including CI and statistical tests would mean doing additional training to make them significative, for instance via bootstraping or multiple iterations of the 5-fold cross-validation. We did not run such experiments for computational resources issues and only provided standard deviation on the results of the 5 folds. If the reviewer has precise ideas of statistical tests that could provide insight on the robustness of the models to multiple trainings or evaluations, we would be pleased to add in our work.
>
> ***3)ensemble method was introduced to reduce performance gap which underscores other models presented; more detail on how models were ensembled.***
>
> The ensembling method used here is very simple : for each method, the probability scores output by each of the 5 models (one per training fold) are averaged. Then, as for a single model, argmax function was used to obtain the final segmentation masks on which the performances are computed.
>
> ***4)Why is 2d performing better than 3d is there a justification? “Surprisingly, the 3D supervised DynUNet is still outperformed by 2D models in term of sensitivity at 1 FP, including by models trained with weak labels.” What is the significance of introducing ensemble method, how were the models ensembled?***
>
> Although it is difficult to rationalize why this happens, we have identified a point that might be part of the explanation. The 3D models are more cautious and make less predictions than 2D models. Consequently, the sensitivity at 1FP that we report for them are often their maximum sensitivity, achieved at a FP per patient rate lower than 1 (we have added to Table 2 an indicator to flag when this is the case), whereas this almost never happens for 2D models.
> This phenomenon can be mitigated with a loss function, that biased the model toward making more predictions, such as Tversky loss, as employed by the winning model of the PI-CAI challenge. Note that this may results in a lower precision of detections. The strategy is particularly effective when different models are subsequently ensembled.

---

### Official Review · Reviewer_heuH · 2024-02-29

**Confidence:** 4
**Preliminary Rating:** 3
**Recommendation:** Poster
**Final Rating:** 4

**Summary:**

The paper proposes to validate a popular size constraint loss proposed by (Kervadec et al., 2018) on large-scale databases of prostate cancer in a weakly supervised setting. Their extensive experiments show that for both in and out-of-distribution data model trained with weak supervision performs on par with the full supervision setting. They also show that the ensemble of these networks always gives better performance compared to a single model.

**Strengths:**

* The paper is well written with clear details about size constraint-based loss.
* Experiments on three different datasets are commendable.
* Results on the out-of-distribution datasets show the generalizability of the method.

**Weaknesses:**

* The paper seems to be a direct application of Kervadec et al., 2018 on a prostate cancer dataset. The authors may want to highlight what is different in their paper compared to Kervadec et al., 2018 or state clearly that their work is focused on validating Kervadec et al., 2018 on prostate cancer datasets.
* The paper is missing some relevant references [1] [2] [3] which are related to weakly supervised settings and Kervadec et al., 2018.
* Authors mention that their full supervision model compares well against the PI-CAI challenge leaderboard. Maybe the authors want to clearly state the best performing method on the leaderboard and report those results for the sack of completeness.
* Image Tag (IT) has been used in a weakly supervised setting. However, details or a citation about the same are missing from the paper. Could the authors please give more information about how IT has been used in a weakly supervised setting?
* Section 2.3 mentions Figure 3 for examples of weak annotations, but Figure 3 doesn't provide that as it shows ground truth and predicted segmentation for different methods. The authors may want to add weak annotations in Figure 3.
* Authors mention that parameters for size constraint-based loss (a and b) were decided empirically. Looking at the histogram of Figure 4 in the appendix, clearly shows that the finalized values of a and b divide the histogram into 3 separate Gaussian distributions. It would be interesting to report model performance for these three different bins of lesion size. I am assuming that the performance would be drastically different for these.
* It is not clear why the authors employed CE + Dice for the full supervision setting and not only CE. It might be a good idea to only use CE in the full supervision setting as it would be easier to compare against other methods.
* Why in the 2D case weakly supervised setting outperform the fully supervised setting and not in the case of 3D? Can authors comment on that?

[1] Tardy, M. and Mateus, D., 2021. Looking for abnormalities in mammograms with self-and weakly supervised reconstruction. IEEE Transactions on Medical Imaging, 40(10), pp.2711-2722.

[2] El Jurdi, R., Petitjean, C., Honeine, P., Cheplygina, V. and Abdallah, F., 2021. High-level prior-based loss functions for medical image segmentation: A survey. Computer Vision and Image Understanding, 210, p.103248.

[3] Bateson, M., Dolz, J., Kervadec, H., Lombaert, H. and Ayed, I.B., 2021. Constrained domain adaptation for image segmentation. IEEE Transactions on Medical Imaging, 40(7), pp.1875-1887.

**Detailed Comments:**

* Usually, I prefer graphs. But I think, in this paper, the graphs are giving too much information (Fig 1 and Fig 2) as there is too much to unpack. Maybe the authors may want to convert it into a table for ease of reading. It would be beneficial if they could provide the table for the same in the appendix.
* Section 3.2 mentions Figure 9. But I am assuming that the authors wanted to point to Figure 2. They may want to correct this typo.
* Section 3.1, Para-2: "This performance compares well the best achievable reported metric on the PI-CAI challenge leader-board" is missing "against".

**Justification Of Final Rating:**

The authors provided a detailed rebuttal.  My main initial concern was regarding the focus of the work. In the revised manuscript, author clearly mentioned that the focus of the work is applying kervadec et al. 2018 method on challenging prostate lesion detection task. Considering that the main focus of the conference is not on novelty but rather on quality of the paper, I am leaning towards weak accept.

**Justification Of The Preliminary Rating:**

The paper seems to be a validation paper that applies Kervadec et al., 2018 on prostate cancer segmentation datasets. As I mentioned in the weakness section, it would be good if the authors mentioned this, as in that case, I would not look for "the novelty" factor while evaluating the work. Also, some things are not clear as mentioned in the weakness and detailed comment section. If the authors clarify this, then I would be happy to accept the paper.

**Questions To Address In The Rebuttal:**

Please address as many questions raised in the weakness and the detailed comment section as you can.

---

> ### Author Response · Authors · 2024-03-18
> **Response to reviewer heuH**
>
> Thank you for your interest in our work and your comments, Reviewer heuH.
>
> We are glad that you appreciate that we provided generalization results. we have corrected the typos you pointed out and added the references you indicated that were missing from the related works. We hope that we will address your main concerns hereinbelow.
>
> ***1) The paper seems to be a direct application of Kervadec et al., 2018 on a prostate cancer dataset. The authors may want to highlight what is different in their paper compared to Kervadec et al., 2018 or state clearly that their work is focused on validating Kervadec et al., 2018 on prostate cancer datasets.***
>
> We would like to thank the reviewer for highlighting this point, which lacked clarity in the paper. The common bounds loss that we use in the paper was already introduced by (Kervadec et al. 2018), for the prostate segmentation task. In (Duran et al., 2022). only the image-tag loss has been adapted to prostate cancer detection. In this work, we validate the use of more precise size constraint (CB loss) on prostate cancer detection task.
>
> Even though this works does not introduce any methodological novelty with regards to the size constraint loss proposed by Kervadec et al, we believe that it still provides a valuable and novel contribution first because it expands the application of this proposed general loss to a more difficult task than binary prostate segmentation illustrated in Kervadec et al. The detection of prostate cancer is indeed more challenging, because the objects to detect are intrinsically more subtle. We demonstrate the positive impact of the CB loss in this situation where the prostate lesion sizes are much less steadily distributed than the prostate ones. Second, we show that the weak learning paradigm of Kervadec allows very promising performance with regards to supervised learning settings.
>
> Finally, we would like to emphasize that our study demonstrates the value of this model for an application of primary interest as recently demonstrated by the great success of the PI-CAI challenge within the urology community.
>
> ***2) Authors mention that their full supervision model compares well against the PI-CAI challenge leaderboard. Maybe the authors want to clearly state the best performing method on the leaderboard and report those results for the sack of completeness.***
>
> The models referred to in this statement can be found here : https://pi-cai.grand-challenge.org/evaluation/open-development-phase/leaderboard/. Of the three sections, the one that is more comparable to the experiments we have carried out is 'Open Development Phase - Tuning Leaderboard', as it is an evaluation on 100 patients from the same centers as the training dataset, hence providing a smaller feature and label shifts. However, we have chosen not to discuss further the comparison of the performance of our reference supervised models with those of the PI-CAI leaderboard for several reasons:
> - The best teams, including the one that won the challenge, mainly leverage ensemble of different models (typically nnUNet and nnDetection) to make more robust predictions and thus improve their AUROC and AP. Moreover, they all combine their lesion segmentation predictions with the knownledge of PSA density, for instance with a logistic regression. Although such ensemble models are commonplace at the top end of any segmentation challenge rankings, we find it of limited interest to compare with them. The models that make the most sense to compare with are those whose performances have been shared by the challenge organizers, i.e. by the users *joeran.bosman* and *anindo* on the leaderboard. Their scores (i.e. mean between AUROC and AP) are as follows : 0.576 for UNet, 0.597 for nnUNet and 0.502 for nnDetection, which are in par with performance achieved by our model (0.62, see Figure 1 and Table 2).
>
> - The data on which the models are evaluated are not the same. We evaluate in a 5-fold cross-validation setup on 1295, whereas the performances reported were obtained after training on all these 1295 patients and testing on 100 new patients, for which we do not have the exams. On these 100 patients, the final predictions are obtained as an ensemble of 25 models (the 5 best models of the 5 training folds), while on PI-CAI, we only report result of single models.
>
> - The models are trained with T2w, ADC and DWI as input modalities while we only used the first two.
>
> With all these elements, we hope that the reviewer has a better idea of why we are claiming that our 3D supervsied baseline model compares well against the PI-CAI challenge leaderboard. We haven't had time to share our model for evaluating them on the hidden cohorts, but we plan to do so in an extension of our work, to provide an even more relevant evaluation.

---

> ### Author Response · Authors · 2024-03-18
> **Response to Reviewer heuH**
>
> ***3)Authors mention that parameters for size constraint-based loss (a and b) were decided empirically. Looking at the histogram of Figure 4 in the appendix, clearly shows that the finalized values of a and b divide the histogram into 3 separate Gaussian distributions. It would be interesting to report model performance for these three different bins of lesion size. I am assuming that the performance would be drastically different for these.***
>
> With the other experiments we had to run, we didn't have time to get the final results on this point. Preliminary results show that sensitivity is indeed much lower for small lesions (volume less than 80 voxels). We are currently investigating this point in more detail to see if this effect is less present on certain models.
>
> ***4)It is not clear why the authors employed CE + Dice for the full supervision setting and not only CE. It might be a good idea to only use CE in the full supervision setting as it would be easier to compare against other methods.***
>
> We are looking at the problem of prostate cancer detection from the point of view of how the annotations are obtained. Our aim is to show that it is possible to compete with fully supervised models using annotations that are quicker to obtain by adding an a priori to the model. With this in mind, we think that it has more sense to compare the weak models with strong supervised baselines (both 2D and 3D). From our experiments, both 2D and 3D supervised models achieve better performance using a combination of CE and generalized DICE loss, that is why we decided to include these results.
>
> As an extension of this work, we plan on testing the CB loss with a nnUNet. However, please note that even though nnUNet is often considered as a state-of-the-art model in medical imaging segmentation, a well-tuned UNet often achieves similar performances. For instance, on the PI-CAI leaderboard, they are 3 comparison databases and 2 training setup (supervised or semi-supervised), and on these 3*3 = 6 possible comparisons, the mean performance of UNet (0.64) is close to that of nnUNet (0.69).
>
> ***5)Why in the 2D case weakly supervised setting outperform the fully supervised setting and not in the case of 3D? Can authors comment on that?***
>
> We believe that the explanation of this phenomenon is two-fold : the overperforming of 3D supervised compared to 2D counterparts *and* the underperforming of 3D weak models compared to 2D counterparts. For the first point, it is not really surprising and in most cases, 3D models benefit from larger context, thus achieving better results than 2D models. For the second point, we hypothesize that it is mainly due by the fact that size bounds are less informative as the number of dimensions increases. Indeed, the lesion size distribution is more spread-out/has more variance in 3D than in 2D, in particular due to the high anisotropy of the images, so the bounds a and b are looser.
>
> ***6)Usually, I prefer graphs. But I think, in this paper, the graphs are giving too much information (Fig 1 and Fig 2) as there is too much to unpack. Maybe the authors may want to convert it into a table for ease of reading. It would be beneficial if they could provide the table for the same in the appendix.***
>
> Even if it is a little dense, we think that the figure allows a better visualization of the results and comparison between the models, so have decided to keep it like this for the main text. However, we do agree that a table providing the numerical results is a valuable addition to the paper, so, as suggested by the reviewer, we added it in the appendix.

---

> ### Comment · Reviewer_heuH · 2024-03-19
> **Response to Rebuttal**
>
> Thank you for providing detailed feedback:
>
> 1) Can you please clearly state at the end of the introduction in the contribution section that the major contribution of your work is to evaluate the Kervadec et al. 2018 method to difficult lesion segmentation task?
>
> 3) I agree that limited time for the rebuttal period might not allow you to perform this analysis, but I would argue that this would have been the easiest. This analysis doesn't require you to perform any new experiments and only extends the performed analysis by dividing the test set into three different sets based on lesion size. The analysis is important as the majority of the lesions' size is greater than the chosen value of b (Fig 4 and Fig 5). In this case, it is important to analyze if the proposed method indeed works well for all lesion sizes, or if it introduces some type of bias based on lesion load.
>
> 4) It is indeed encouraging that choosing CE+Dice over CE was based on empirical performance and the authors chose the strongest option. However, this raises the question of why CB losses were only employed with CE and not CE+Dice. As it would make a better case of showing the usefulness of CB losses.

---

> > ### Author Response · Authors · 2024-03-26
> >
> > 1) We have updated the contribution section and the abstract to make our main contribution clearer. We hope that this new version will better meet your expectations.
> >
> > 2) We would like to apologize as there was a mistake on the distributions of the lesion sizes that we had displayed in the appendix. The main peak/bin represented in fact the size of the prostate, which explains its higher frequency compared to lesion and its largely superior mean size. We have corrected this mistake with the true distribution, which does not contain anymore the three main bins of lesion sizes that you mentionned in your review. However, to address the point raised by the reviewer concerning the differences in performance depending on the size of the lesions to be detected, we provide hereunder a table with the sensitivity at 1 FP per patient for the best weakly supervised and fully supervised models for three bins of sizes. P0-P10 is for lesions between under 10th percentile in size distribution (*small lesions*), P10-90 is for lesions between the 10th and 90th percentile (*normal size lesions*), and P90-P100 is for the lesions above the 90th percentile (*large lesions*). As expected, we observe some discrepancies in performance depending on the size of the lesion to be detected, smaller lesions being generally less well detected by both models. Moreover, the weakly supervised model, even though it sets lower and upper bounds on the object sizes to be segmented, which one intuits could have biased its performance towards underperforming on small and/or big lesions, does not seem to be more biased than the baseline 3D supervised baseline. Please let us think if you believe that this analysis on the performance depending on the lesion size is of interest to improve the quality of the paper.
> >
> > |   PI-CAI                        | P0-P10 | P10-P90   | P90-P100 |
> > |--------------------------------|---------------|------|-------|
> > | 2D + CE + IT + CB              | 0.391       |  0.745    | 0.913     |
> > | 3D Supervised                 |  0.348          | 0.610     | 0.826     |
> >
> > |   Private dataset                       | P0-P10 | P10-P90   | P90-P100 |
> > |--------------------------------|---------------|------|-------|
> > | 2D + CE + IT + CB              | 0.264      | 0.404    | 0.261     |
> > | 3D Supervised                 | 0.118     | 0.323    | 0.339     |
> >
> > |   Prostate158                       | P0-P10 | P10-P90   | P90-P100 |
> > |--------------------------------|---------------|------|-------|
> > | 2D + CE + IT + CB              | 0.733 | 0.526   | 0.667    |
> > | 3D Supervised        | 0.444    | 0.410    | 0.689     |
> >
> >
> > 3) DICE loss is particularly well suited for cases where there is a disparity between the sizes of the object that are aimed to be segmented. Indeed, it penalizes a model based on the proportion of ground truth that is well segmented, therefore handling better class imbalance than cross-entropy. In our weakly supervised setup, and given how we obtain the weak annotations (point scribles), most of the objects have the exact same size, so we intuited that the model would not benefit from being trained with cross-entropy and DICE loss. We trained the best performing weakly supervised model, namely 2D CE+IT+CB, with additional DICE loss and provide in the table below its performance. It gives slightly better results overall on most metrics and databases. However, as the discussion period is coming to its end, we do not have time to train all the weakly supervised models, select the best hyperparameters and evaluate them with the same protocol as that of all the other experiments. Thus, we can not include the results below in the article but hope that it will convince the reviewer of the value of applying Kervadec's loss to this more challenging prostate cancer detection task.
> >
> > |   PI-CAI                        | Sensi at 1 FP | AP   | AUROC |
> > |--------------------------------|---------------|------|-------|
> > | 2D + CE + IT + CB              | 0.746 +/- 0.028       |  0.256 +/- 0.042    | 0.776 +/- 019     |
> > | 2D + CE + IT + CB + DICE                 |  0.783 +/- 0.059          | 0.268 +/- 0.080     | 0.790 +/- 0.017     |
> >
> > |   Private dataset                       | Sensi at 1 FP | AP   | AUROC |
> > |--------------------------------|---------------|------|-------|
> > | 2D + CE + IT + CB              | 0.376 +/- 0.056 (0.401)      | 0.280 +/- 0.047 (0.399)    | 0.641 +/- 0.031 (0.661)     |
> > | 32D + CE + IT + CB + DICE                 | 0.404 +/- 0.037 (0.457)     | 0.278 +/- 0.032 (0.414)    | 0.776 +/- 0.019 (0.686)     |
> >
> > |   Prostate158                       | Sensi at 1 FP | AP   | AUROC |
> > |--------------------------------|---------------|------|-------|
> > | 2D + CE + IT + CB              | 0.542 +/- 0.059 (0.635)  | 0.388 +/- 0.041 (0.421)   | 0.726 +/- 0.028 (0.781)    |
> > | 2D + CE + IT + CB + DICE        | 0.577 +/- 0.051 (0.635)     | 0.377 +/- 0.030 (0.500)    | 0.733 +/- 0.017 (0.786)     |

---

> > > ### Comment · Reviewer_heuH · 2024-03-26
> > > **Final response**
> > >
> > > Thank you for the detailed feedback.
> > >
> > > 1. Thank you for the provided analysis and correcting the histogram figure. From the revised figure-5 I can see that choice of b is still really small and majority of 2D lesions fall outside the range of  a-b.  Considering that, I am not sure why the provided analysis was down for P0-P10, P10-P90, and P90-P100. These percentiles were not used to select the hyperparameters. As per my original comment, the analysis would be more useful for 0-a, a-b, and b-max(lesion_size). Nonetheless, provided analysis indeed shows that there is indeed a bias. Especially, for the small lesion size, but this is expected. However, I was surprised that peformance was higher for the large lesion. But maybe this is due to the fact that the paper measures on detection performance and not segmentation. In the future, authors may want to show this as one of the features of their work that lesion detection performance, especially for the large lesions are not affected by the choice of hyperparameter b. Also, in the future authors may want to lesion wise dice scores in the table, and also number of false positive lesions detected per patient in the performance evaluation.
> > >
> > > 2. Thank you the provided analysis. Indeed, as I was expecting, the performance for weakly supervised methods indeed seems to improve with addition of Dice. I would encourage the authors to either stick to only CE or CE+Dice for both supervised and weakly supervised methods (the choice is theirs). As it allows better comparison.
> > >
> > > Considering all the efforts by the authors, I am converting my rating to weak acceptance

---

> > > > ### Author Response · Authors · 2024-03-28
> > > > **Final response**
> > > >
> > > > 1) Providing analysis for P0-P10, P10-90 and P90-100 was an easier way to discuss the bias with the lesion sizes, and we think that finer 2D analysis for 0-a, a-b and b-max(lesion size) would yield similar conclusions. We agree with you that the higher performance for large lesion if indeed due to the rather loose criteria for counting true positives. Extending the analysis to the influence of the hyperparameters on performance by lesion size is indeed a necesseray perspective of this paper, to assess the robustness of Kervadec's method on this task.  As a final addition, we have added to Appendix D the maximum sensitivity and average false positive per patient. It may help future readers to better understand the behaviour of the models, such as the difference between 2D and 3D models, or the impact of ensembling on the number of false positives.
> > > >
> > > > 2) Given the short amount of time before the closure of the discussion phase, we do not have time to train the weakly supervised model with CE + DICE (and redo the grid search) or to train and evalute the baselines with CE loss only (although we agree that it would have allowed for better comparison) and we will leave the paper in its current state.
> > > >
> > > >
> > > > It would be our pleasure to continue the discussion about the choice of cost function and size constraint bounds at the conference if the paper is accepted. We would like to thank you once again for your interest in our work and for taking the time to exchange with us.

---

### Author Response · Authors · 2024-03-18
**Comments and response to all reviewers**

Dear Reviewers,

We first would like to thank you all for the time spent on reviewing this paper and for your fruitful feedbacks. We have tried to address most of them in the following responses and append the manuscript as far as we could due to time and space constraints. We hope that the quality of the manuscript will meet your expectations. The main comments, some of them shared by the reviewers, are the followings:

- Clarify the link between this work and those of (Kervadec et al., 2018) and (Duran et al., 2022).

- Add an appendix in which we compare several ways of obtaining weak labels.

- We had forgotten to show the weak annotations in figures 3 and 8; this has been corrected. There was also a mistake in Figure 1, where the performance of the model 'Reference' was underevaluated (instead, we were reporting the performance of the ensemble 2D supervised model). We would like to apologize for this error, which does not change most of the analysis made in our work.

- Provide explanation of the differences in performances between certain models. In particular, please see answer to point 4) of reviewer 49yd and point 3) of reviewer aYxy.

Changes in the paper have been highlighted in red to facilitate the reviewers' revision.

To expand our comparison with concurrent models, we are currently running a 5-fold training of nnDetection (Baumgartner et al., 2022), a state-of-the-art model in medical imaging object detection. The results will be soon added to the paper. For now, we can report on this model's performance on : AP = 0.265 and AUROC = 0.735. Please note that the model uses a different kind of weak supervision than we do, namely bounding boxes, and we intuit than this is a stronger supervision when evaluation is performed with detection metrics. Please also read the reply to reviewer heuH for more details on how PI-CAI-s leaderboard results compare to our own.

We'll be pleased to have feedbacks from the reviewers and exchange with them during the discussion phase to further improve the quality of the paper.

---

> ### Author Response · Authors · 2024-03-26
>
> Dears Reviewers,
>
> As requested, we have extended our comparison to state-of-the-art models by adding nnDetection to it. nnDetection is a 3D object detection network that uses segmentation masks as supervision. To compare it with our models, *i.e.* in a weakly supervised setting, we only trained using bounding boxes by replacing full segmentations by bounding boxes segmentations. We considered two setups : a first using the real segmentations to create the bounding boxes (*whole lesions BB*) and a second where the boxes were created from the dotted annotations that we used for other weakly supervised models (*weak lesions BB*). We have trained and evaluated it on the same setup as all the other models.
>
> We provide hereunder the results of these two additional models (and show in the table the best weakly supervised model on dot annotations and the best supervised model for comparison). We will add these results to the table, figures and discussion very soon in a new version of the article.
>
> |   PI-CAI                        | Sensi at 1 FP | AP   | AUROC |
> |--------------------------------|---------------|------|-------|
> | 2D + CE + IT + CB              | 0.746 +/- 0.028       |  0.256 +/- 0.042    | 0.776 +/- 019     |
> | 3D Supervised                  |  0.705 +/- 0.058          | 0.412 +/- 0.047     | 0.825 +/- 0.013     |
> | nnDetection (whole lesion BB)  |  0.476 +/- 0.042      | 0.276 +/- 0.043    | 0.758 +/- 0.034     |
> | nnDetection (weak lesions BB)  | 0.327 +/- 0.057        | 0.189 +/- 0.019    | 0.746 +/- 0.044    |
>
> |   Private dataset                       | Sensi at 1 FP | AP   | AUROC |
> |--------------------------------|---------------|------|-------|
> | 2D + CE + IT + CB              | 0.376 +/- 0.056 (0.401)      | 0.280 +/- 0.047 (0.399)    | 0.641 +/- 0.031 (0.661)     |
> | 3D Supervised                  | 0.304 +/- 0.045 (0.272)     | 0.337 +/- 0.030 (0.432)    | 0.627 +/- 0.037 (0.626)     |
> | nnDetection (whole lesion BB)  | 0.332 +/- 0.045 (0.365)     | 0.240 +/- 0.059 (0.287)    | 0.614 +/- 0.045 (0.626)     |
> | nnDetection (weak lesions BB)  | 0.215 +/- 0.025 (0.267)     | 0.122 +/- 0.014 (0.165)  | 0.602 +/- 0.025 (0.633)     |
>
> |   Prostate158                       | Sensi at 1 FP | AP   | AUROC |
> |--------------------------------|---------------|------|-------|
> | 2D + CE + IT + CB              | 0.542 +/- 0.059 (0.635)  | 0.388 +/- 0.041 (0.421)   | 0.726 +/- 0.028 (0.781)    |
> | 3D Supervised                  | 0.438 +/- 0.071 (0.438)     | 0.366 +/- 0.031 (0.484)    | 0.733 +/- 0.019 (0.780)     |
> | nnDetection (whole lesion BB)  | 0.295 +/- 0.039 (0.322)    | 0.199 +/- 0.054 (0.237)   | 0.632 +/- 0.042 (0.673)     |
> | nnDetection (weak lesions BB)  | 0.238 +/- 0.052 (0.311)   | 0.140 +/- 0.047 (0.197)    | 0.619 +/- 0.073 (0.668)     |

---

### Meta-Review · Area_Chair_M7HE · 2024-04-04

**Recommendation:** Accept (Poster)
**Confidence:** 4

**Metareview:**

The Meta-Reviewer has read all reviews and rebuttal of authors.

The work has received mixed reviews. There is a consensus that the main value of the paper is rather on the evaluation of an existing approach for prostate cancer detection, rather than contributing novel technical methodology. It is acknowledged by reviewers that the paper is well written, the evaluation is performed on 3 databases (which is commendable) and that the experimental results provide some useful insights, eventhough there are limitations (eg further baselines/comparisons could have been performed).

Overall, the consensus between reviewers and this meta-reviewer is that the paper provides sufficient experimental insights to be of interest to the community. Therefore I will recommend accepting the paper

---

### Decision · Program_Chairs · 2024-04-05

Accept (Poster)